# Remote sensing of salmonid spawning sites in freshwater ecosystems: The potential of low-cost UAV data

Lieke Ponsioen *, Kalina H. Kapralova¤, Fredrik Holm, Benjamin D. Hennig

Institute of Life and Environmental Sciences, University of Iceland, Reykjavík, Iceland

¤ Current address: Institute for Experimental Pathology at Keldur, University of Iceland, Reykjavík, Iceland
* lieke@hi.is

## Abstract

Salmonids are especially vulnerable during their embryonic development, but monitoring of their spawning grounds is rare and often relies on manual counting of their nests (redds). This method, however, is prone to sampling errors resulting in over- or underestimations of redd counts. Salmonid spawning habitat in shallow water areas can be distinguished by their visible reflection which makes the use of standard unmanned aerial vehicles (UAV) a viable option for their mapping. Here, we aimed to develop a standardised approach to detect salmonid spawning habitat that is easy and low-cost. We used a semi-automated approach by applying supervised classification techniques to UAV derived RGB imagery from two contrasting lakes in Iceland. For both lakes six endmember classes were obtained with high accuracies. Most importantly, producer's and user's accuracy for classifying spawning redds was >90% after applying post-classification improvements for both study areas. What we are proposing here is an entirely new approach for monitoring spawning habitats which will address some the major shortcomings of the widely used redd count method e.g. collecting and analysing large amounts of data cost and time efficiently, limiting observer bias, and allowing for precise quantification over different temporal and spatial scales.

## Introduction

Salmonids encompass a diverse group of cold-temperate fish and hold an important role culturally [1, 2] and economically as a fisheries resource [3, 4]. They also play a key-role in shaping various ecological processes crucial to the functioning and health of freshwater and marine environments [5–7]. However, many salmonid species have been in decline due to anthropogenic stressors with climate change a major driver [8–12]. Salmonids are especially vulnerable during their embryonic development [13], relying on highly oxygenated water flow and low temperatures. Thus, rising temperatures [8] and other anthropogenic stressors such as human disturbance [14] present a major threat to their survival. This makes identifying and mapping the breeding and nursery grounds of these species important for their conservation.

**Funding:** Parts of this research were undertaken with funding provided by the University of Iceland Research Fund received by BDH. The funders had no role in study design, data collection and analysis, decision to publish, or preparation of the manuscript.

**Competing interests:** The authors have declared that no competing interests exist.

Nursery grounds are habitats which contribute disproportionally to the size and numbers of adults [15] and information on relative abundance and maturity status of fish, as well as information on their geographical and ecological characteristics can be used to identify them [16, 17]. Most salmonid species are characterised as gravel nests spawners [18] and spawning redds (*i.e.* spawning nests) appear as irregular or regular shapes that contrast with the undisturbed area surrounding them [19], making them visible from above. Spawning redds provide an important source of information for management purposes, including monitoring population size and estimating carrying capacity of spawning habitats [20]. However, the monitoring of their spawning grounds is rare and often relies on manual counting of the redds [20–23]. This method is prone to sampling errors resulting in over- and/or underestimations of redd numbers with observer counts ranging from 28% to 254% [24, 25]. Sampling error can be assigned to a wide variety of factors, such as, low visibility due to physical characteristics of the spawning redd location (*e.g.* water depth, substrate composition), variation in redds (*e.g.* redd size, superimposition), incomplete sampling of spawning areas in space and/or time, and inexperienced observers, among others [24, 26]. Apart from sampling errors, manual redd counting is a time-consuming, it can be unsafe and at times difficult, and it does not allow for precise quantification over different temporal and/or spatial scales [24].

The use of unmanned aerial vehicles (UAVs), also known as drones, for detecting salmonid redds is being explored to address some of the shortcomings [27]. UAVs have already shown their potential in supporting ecological fieldwork by providing high-frequency, high-quality, and low-cost data [28–30]. For example, the use of UAVs have been essential in monitoring breeding colonies of seabird populations [31], characterising sensitive habitats such as juvenile fish nursery grounds [29], and supporting monitoring of salmonid spawning nests [21]. Despite the research on using UAVs to detect salmonid redds, there is still a great need for developing an easy, standardised, and low-cost approach to map entire spawning habitats as current methods either still rely on manual counting [27, 32], utilise rather complex methodology [33] or need specialised equipment [34].

Here, we created a pipeline using a semi-automated remote sensing approach by applying pixel-based image classification techniques to RGB imagery derived from an UAV, which was tested at two environmentally contrasting Icelandic lakes, lake Thingvallavatn and lake Ellidavatn. Lake Thingvallavatn is characterised by contrasting features and heterogenous topography, while lake Ellidavatn presents a more homogeneous ground level with low contrast between spawning redds and the surrounding underwater vegetation. Rather than estimating the number of spawning redds, the method described here maps the spawning redds visible on a RGB image allowing for precise quantification of their size over different temporal and spatial scales and it helps limiting sampling errors due to observer bias and/or superimposition.

## Material and methods

### Study areas

Two study areas (*i.e.* lake Thingvallavatn, lake Ellidavatn) with contrasting environmental characteristics were selected within the Icelandic freshwater ecosystems to establish and validate the method presented here.

The method was first established in lake Thingvallavatn, located in southwestern Iceland. In this study we focussed on the well-studied spawning grounds of Ólafsdráttur, an area of the lake located within the protected area of Thingvellir national park, which is known to host the spawning of the large benthic Arctic charr (*Salvelinus alpinus* Linnaeus, 1758) in July and August each year. The spawning redds in this area are located in both very shallow (*i.e.* 0.5–1.5 m) and deeper water (*i.e.* 1.5–5 m). The second study area, lake Ellidavatn, was used to validate

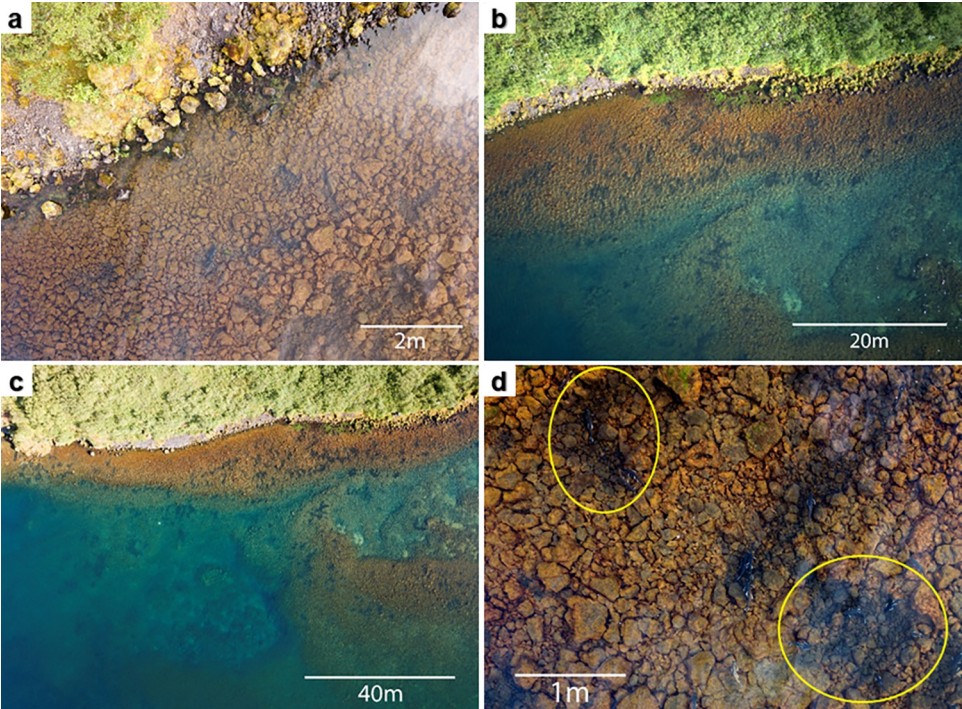

**Fig 1. Aerial images spawning grounds lake Thingvallavatn.** (a-c): Aerial images of the lake Thingvallavatn spawning grounds at different altitudes: 10 m (a), 50 m (b), and 100 m (c). (d): close-up view of the dark spawning redds that are made by Arctic charr. Two examples of redds have been marked with yellow circles.

the method. The lake is situated in the urban area of Reykjavík. Here we focussed on Arctic charr spawning grounds located on the northern shore of the lake in shallow water (± 1 m) where they spawn from September to November.

The study is based on the assumption that salmonid redd structures in shallow water areas can be distinguished by their visible reflection which makes the use of standard UAV a viable option. Arctic charr females exhibit particular behaviour by cleaning rocks from debris, silt, and algae to prepare the spots for spawning. Due to this behaviour redds can be identified from the air as dark areas of gravel and rocks since the cleared debris, silt, and algae have a lighter colouring than the rocks themselves (Fig 1D). The distinction between spawning redds and other geographical features depends on the surrounding environment. The two study areas chosen in this study represent contrasting environments. Lake Thingvallavatn is characterised by well-contrasting features and variance in deep water level, while lake Ellidavatn presents a more homogeneous ground level with low contrast between spawning redds and the surrounding underwater vegetation making the distinction of the redds more challenging.

## Data acquisition

In preparation to data acquisition the occurrence of spawning redds was verified in lake Thingvallavatn. This was based on the following two steps: (i) locating spawning redds from the shore, and (ii) verifying that the observed redds are used for spawning through in-situ observations and video recordings of spawning activity [35], and noting the presence eggs to confirm successful spawning. Data collection was undertaken using a DJI Mavic Pro drone in lake Thingvallavatn and a DJI Mavic 2 Zoom drone in lake Ellidavatn (S1 Table) equipped with a RGB colour camera and a remote controller to command the drone. For lake

Thingvallavatn automatic camera settings were used with a polarising filter, while in lake Ellidavatn manual camera settings set to the lowest ISO and a shutter speed of 1/30 s was used without a polarising filter. The remote controller was operated on an Android smartphone with the DJI GO 4 application developed by the UAV manufacturer. On both days of data acquisition ground truth reference data was collected by visually confirming the presence of spawning redds from the shore just before the UAV survey. To avoid the sun's reflection on the water surface, the aerial surveys were taken before sunset, and in minimal wind conditions to avoid ripples in the water surface. The aerial surveys were completed on 27-Jul-2018, between 17:00h and 19:00h in lake Thingvallavatn (after flight permission was obtained from National Park Thingvellir), and on 3-Nov-2022, between 11:30h and 11:45h in lake Ellidavatn. The timing of the survey was chosen in consideration with the spawning season of the Arctic charr and optimal weather conditions. Recorded wind speed in the lake Thingvallavatn survey area was 2 m/s wind SSE direction and at lake Ellidavatn wind speed was 3 m/s from SW direction during the survey. The total duration of the lake Thingvallavatn survey was one full hour (three full batteries), narrowing the aerial time but obtaining sufficient coverage of the study area. Over the flight course multiple images were taken but in this study only a single image per site was selected and used for data processing. For lake Ellidavatn a fourth of the time was needed due to a smaller area covered by the sampling site, here too multiple images were taken but only a single image was used for the analysis. In both cases the selected images covered the ground truth reference sites through which the presence of the spawning areas could be confirmed. The flight direction of the surveys was controlled manually by an experienced drone pilot. It followed the shallow shoreline (0–10 m) where the spawning redds were more concentrated. The flight path at lake Thingvallavatn was taken over the same areal extent at 10 m, 50 m, and 100 m with the intention of capturing different resolution images for comparison (Fig 1A–1C). The challenges of this methodology lie in the heterogeneous structure of the spawning redds and the similar reflectance with water vegetation and deep water that can interfere with the classification of the features.

## Data processing

The software *ENVI* version 5.1 (Exelis Visual Information Solutions, Boulder, Colorado) was used for processing and classification analysis. For data processing, only images with the least sunlight reflection and wind ripples were selected. Endmember classes were selected by an experienced biologist while taking the ground truth information into account after the images of the spawning ground were taken. For lake Thingvallavatn and lake Ellidavatn six endmember classes were selected based on the ecological and the geographical features at the sites. For lake Thingvallavatn the classes selected included "spawning redds", a dark colour where this feature dominates; "vegetation", a green coloured area located on land; "underwater rocks", visible as a lighter colouring than the spawning redds; "deep water", a deeper, darker blue coloured area with few ripples caused by wind on the water surface; "shoreline", characterised by the lightest colouring; and "surface rocks", occurring over the shoreline and some parts of the shallow water recognisable as yellow colouring (Fig 2A). The endmember classes selected from the lake Ellidavatn image were slightly different due to the presence of anthropogenic features and the ecology of the water bottom. The following selection was made: "spawning redds", a darker colour compared to the lighter background; "vegetation", recognisable by its light brown colouring; "underwater rocks", characterised by brown colour; "aquatic vegetation", a darker colour compared to the spawning redds; "anthropogenic feature", located on land and grey/brown in colour (a road); and "sediment", a green and brown colouring covering most of the lake bottom (Fig 3A).

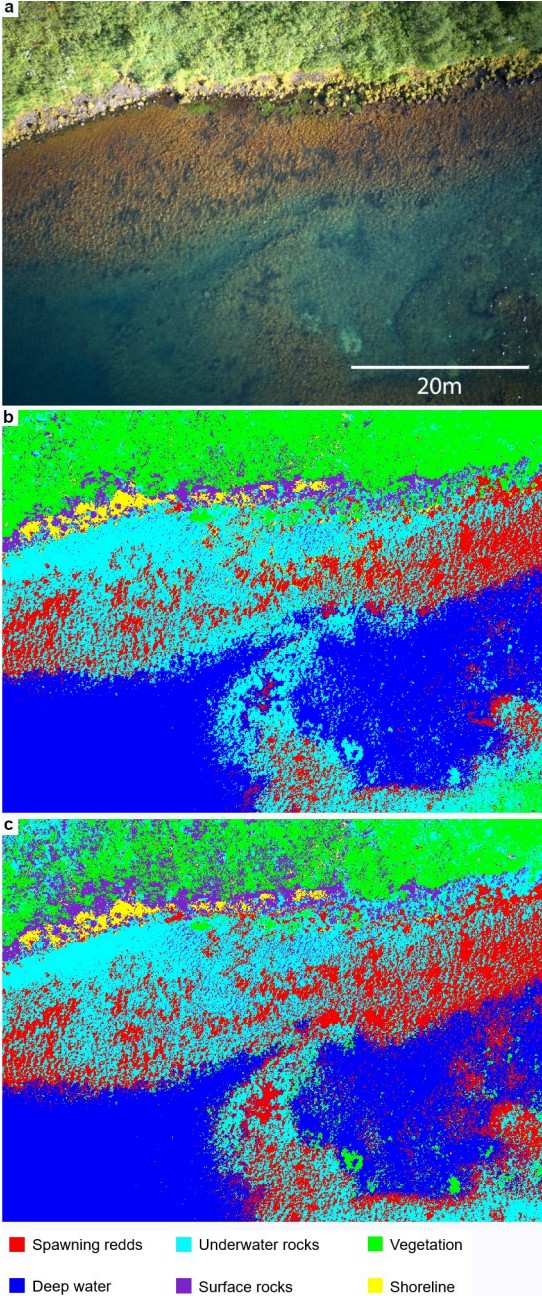

Fig 2. **Endmember classes in lake Thingvallavatn.** (a): RGB image from lake Thingvallavatn. (b): Overview of the endmember classes obtained with the maximum likelihood classification. (c): Overview of the endmember classes obtained with the neural net classification. Classified pixels as spawning redds are in red, vegetation in green, underwater rocks in cyan, deep water in blue, shoreline in yellow, and surface rocks in purple.

Fifteen training samples with an average of 65 pixels each, to not over- or undertrain the classification method, of each spectral class were selected on the images of the spawning grounds to allow for reasonable estimates to determine the mean vector and the covariance matrix [36]. Followed by, a supervised classification method to classify the pixels.

Three post-classification methods (*i.e.* majority-minority analysis, sieve classes method, clump classes method) were applied in order to improve the accuracy by correcting isolated or

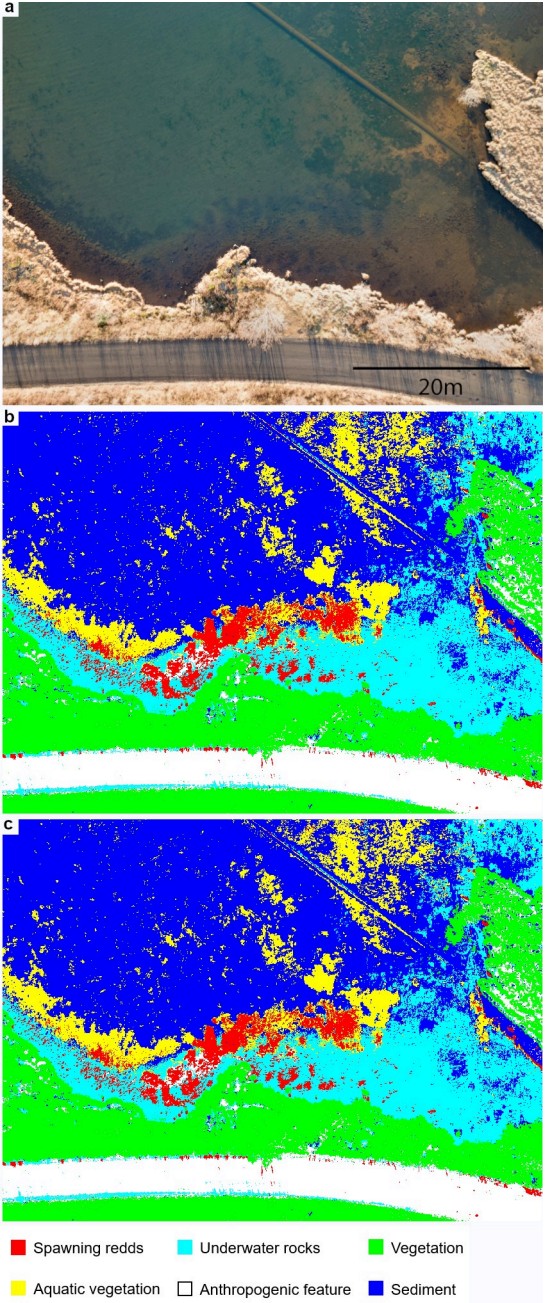

**Fig 3. Endmember classes in lake Ellidavatn.** (a): RGB image from lake Ellidavatn. (b): Overview of the endmember classes obtained with the maximum likelihood classification. (c): Overview of the endmember classes obtained with the neural net classification. Classified pixels as spawning redds are in red, underwater rocks in cyan, vegetation in green, aquatic vegetation in yellow, anthropogenic feature in white, and sediment in blue.

misclassified pixels. First, the classified output was filtered by removing spurious pixels with a majority-minority analysis [37]. This analysis changes spurious or "false" pixels to the class value that the majority of the pixels in the manually indicated kernel belong to. The following parameters were selected: majority for the analysis method, a kernel size of 3, and a centre pixel weight of 1. In addition, isolated pixels were corrected using the sieve classes method [38,

39]. This method looks at neighbouring pixels to determine if a pixel is grouped with the same endmember classes surrounding the pixel. For this study pixel connectivity was set to four and the minimum size to two. Lastly, the clump classes method was applied to clump similarly classified areas together adjacent from each other [38]. Following a visual examination of the initial classification results, the size parameter for this method was set to three.

### Accuracy assessment

To determine the accuracy of our method a confusion matrix using ground truth regions of interest was performed. A ground truth ROI (ROI: regions of interest) dataset was generated by selecting pixels for each spectral endmember class once again (different from the pixels selected as training data). The training dataset was then paired with the ground truth ROIs to determine what percentage of the ROI pixels were or were not contained in one of the endmember classes. The confusion matrix reported overall accuracy of the applied supervised algorithm by expressing the percentage of correctly classified pixels of all endmember classes. Producer's accuracy (PA) is defined as the probability that each endmember class is classified correctly. User's accuracy (UA) is defined as the probability that the classification map represents the ground truth data. Furthermore, the kappa coefficient was used to evaluate the classification accuracy and can be interpreted as a value ranging from 0 to 1 that explains the difference between the observed classification of the endmember classes and the reference data [40].

## Results

The method encompasses three main stages: data acquisition, data processing, and the accuracy assessment (Fig 4).

To compare different image resolutions the flight path at lake Thingvallavatn was taken over the same areal extent at 10 m, 50 m, and 100 m (Fig 1A–1C). Following endmember collection and subsequent analyses, the 10 m and 100 m imagery were discarded. The 10 m imagery was discarded due to the high level of detail the picture provided which made overall

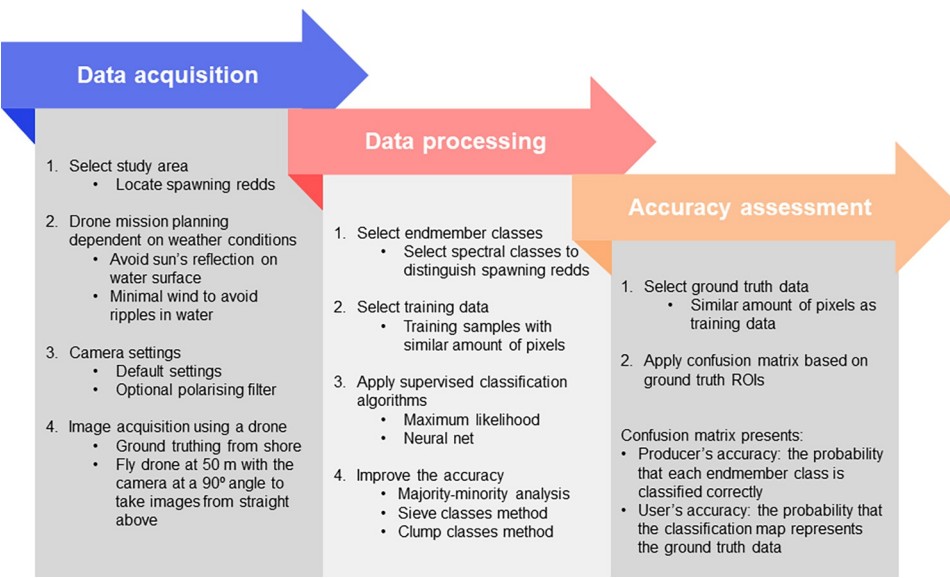

**Fig 4. Main steps of the method to detect spawning grounds.**

patterns of spawning grounds difficult to see, increased the level of processing time, and gave an insufficient number of classes. The 100 m imagery was discarded due to reflectance and low accuracy results. The 50 m imagery was deemed suitable for the classification methods as it provided the optimal contrast and lowest environmental reflectance in relation to the spawning grounds. On the 50 m imagery from lake Thingvallavatn seven common supervised image classification methods were tested for their accuracy at classifying spawning redds (S2 Table, all data files are available from https://github.com/Worldmapper/ecoUAV-iceland). Both the neural net classification (PA: 91.09%, UA: 79.62%) and the maximum likelihood classification (PA: 89.70%, UA: 90.51%) showed promising results at classifying the spawning redds as well as overall accuracy (neural net 83.15%, maximum likelihood 87.44%).

After applying the maximum likelihood and the neural net classification on the training samples of the lake Thingvallavatn area the endmember classes were classified and presented as following (Fig 2B and 2C): Class 1 (spawning redds; in red) is a dominant class covering parts of the shallow area, close to the shore. This class can be described as a number of small, irregular groups of pixels connected by few pixels surrounding the groups. Class 2 (vegetation; in green) covers the entire top part of the image in one group of pixels. Class 3 (underwater rocks; in cyan) is distributed covering the pixels adjacent to class 1 and completes the shallow area. The major difference between these two classes is the lack of irregular groups of pixels in class 3. Class 4 (deep water; in blue) can be described as a very defined and mostly smooth class. Class 5 (shoreline; in yellow) is defined by little groups of pixels adjacent to class 2. This class is very heterogeneous and has few pixels per group that do not cover a large area. Lastly, class 6 (surface rocks; in purple) can be found adjacent to class 2 and is defined by little group of pixels. The endmember classes selected for the lake Ellidavatn study area are presented as following (Fig 3B and 3C): Class 1 (spawning redds; in red) appears in the shallow water and occurs as irregular groups of pixels with connected by few pixels surrounding the groups. Additionally, few irregular groups are located on land. Class 2 (vegetation; in green) is a dominant cover class in the land area and has a high density of groups of pixels connected to each other. Class 3 (underwater rocks; in cyan) is a dominant class covering the shallow area. This class is adjacent to class 1 and 4. Class 4 (aquatic vegetation; in yellow) covers areas towards deeper water. The groups of pixels are of irregular shape. Class 5 (anthropogenic feature; in white) covers a long strip in the bottom of the image. Lastly, class 6 (sediment; in blue) consists out of a high-density main group of pixels covering most of the image.

After applying the maximum likelihood classification for the lake Thingvallavatn study area the accuracy assessment reported a PA of 89.70% and a UA of 90.51% for classifying spawning redds before any post-classification improvements were applied (S3 Table). After applying the post-classification methods these values reported 90.99% for PA and 96.33% for the UA (Table 1). Spawning redds were mostly correctly classified, however, 15 pixels were classified as underwater rocks, 74 pixels as deep water, and 2 pixels as shoreline. The other endmember classes reported >80% accuracy for PA after post-classification methods and >78% for UA (S3 Table). Furthermore, looking at the overall accuracy and kappa coefficient applying post-classification methods improved the accuracy from 87.44% to 90.78% and the kappa coefficient from 0.85 to 0.89 (S3 Table).

The error matrix of the neural net classification reported 92.18% PA and 88.92% UA for the spawning redds (Table 2) which were improved from 91.09% PA and 79.65% UA (S4 Table) after applying post-classification methods. Seventeen pixels were incorrectly classified as underwater rocks and 62 pixels as deep water (Table 2). The PA for the other endmember classes reported >80% except for the vegetation endmember class which only reported 70.31% accuracy. The lowest UA reported 69.13% accuracy for the underwater rocks class while the other endmember classes reported above 88%. Lastly, overall accuracy reported 83.15% before

**Table 1. Error matrix of the maximum likelihood classification in pixels for lake Thingvallavatn.** Producer's accuracy (PA) and user's accuracy (UA) are reported after applying post-classification methods. The columns represent the true classes and the rows represent the classifier's predictions. A single pixel was unclassified due to the post-classification improvement methods.

| Class | Spawning redds | Vegetation | Underwater rocks | Deep water | Shoreline | Surface rocks | Total | UA (%) |
|---|---|---|---|---|---|---|---|---|
| **Unclassified** | 0 | 1 | 0 | 0 | 0 | 0 | 1 | |
| **Spawning redds** | **919** | 0 | 26 | 9 | 0 | 0 | 954 | 96.33 |
| **Vegetation** | 0 | **867** | 39 | 0 | 0 | 70 | 976 | 88.83 |
| **Underwater rocks** | 15 | 92 | **924** | 26 | 0 | 116 | 1173 | 78.77 |
| **Deep water** | 74 | 0 | 24 | **966** | 0 | 0 | 1064 | 90.79 |
| **Shoreline** | 2 | 20 | 0 | 0 | **1011** | 21 | 1054 | 95.92 |
| **Surface rocks** | 0 | 27 | 0 | 0 | 6 | **907** | 940 | 96.49 |
| **Total** | 1010 | 1007 | 1013 | 1001 | 1017 | 1114 | | |
| **PA (%)** | 90.99 | 86.10 | 91.21 | 96.50 | 99.41 | 81.42 | | |

post-classification methods were applied and improved to 86.95% after, and kappa coefficient improved from 0.80 to 0.84 (S4 Table).

For the lake Ellidavatn study area the accuracy assessment of the maximum likelihood classification reported a PA of 91.79% and a UA of 84.67% for classifying spawning redds before post-classification methods were applied (S5 Table) which after applying the post-classification improvements reached an accuracy of 95.15% and 87.06%, respectively. Misclassified pixels were assigned to the aquatic vegetation class (49 pixels) (Table 3). The other endmember classes reported high PA and UA (>75%). Overall accuracy reported 86.88% before post-classification methods and was improved to 88.89% after applying the methods (S5 Table). Additionally, the kappa coefficient reported 0.84 (before post-classification) and 0.87 (after post-classification) (S5 Table).

The assessment of the neural network classification in lake Ellidavatn reported 99.80% PA and 85.58% UA for classifying spawning redds after applying post-classification methods (Table 4). These accuracies were improved from 96.64% PA and 82.03% UA (S6 Table). Misclassified pixels were classified as aquatic vegetation (2 pixels) (Table 4). The vegetation, underwater rocks, and anthropogenic features endmember classes reported high PA (>77%) and UA (>69%) accuracies, however, the aquatic vegetation and sediment endmember classes scored low for PA (<66%) and UA (<70%) (Table 4). Overall accuracy was improved from 79.08% to 80.66% accuracy after applying post-classification methods and the kappa coefficient from 0.75 to 0.77 (S6 Table).

**Table 2. Error matrix of the neural network classification in pixels for lake Thingvallavatn.** Producer's accuracy (PA) and user's accuracy (UA) are reported after applying post-classification methods. The columns represent the true classes and the rows represent the classifier's predictions.

| Class | Spawning redds | Vegetation | Underwater rocks | Deep water | Shoreline | Surface rocks | Total | UA (%) |
|---|---|---|---|---|---|---|---|---|
| **Unclassified** | 0 | 0 | 0 | 0 | 0 | 0 | 0 | |
| **Spawning redds** | **931** | 2 | 23 | 82 | 9 | 0 | 1047 | 88.92 |
| **Vegetation** | 0 | **708** | 29 | 7 | 0 | 25 | 769 | 92.07 |
| **Underwater rocks** | 17 | 213 | **909** | 2 | 0 | 174 | 1315 | 69.13 |
| **Deep water** | 62 | 0 | 16 | **910** | 0 | 0 | 988 | 92.11 |
| **Shoreline** | 0 | 32 | 0 | 0 | **1006** | 21 | 1059 | 95.00 |
| **Surface rocks** | 0 | 52 | 36 | 0 | 2 | **894** | 984 | 90.85 |
| **Total** | 1010 | 1007 | 1013 | 1001 | 1017 | 1114 | | |
| **PA (%)** | 92.18 | 70.31 | 89.73 | 90.91 | 98.92 | 80.25 | | |

**Table 3. Error matrix of the maximum likelihood classification in pixels for lake Ellidavatn.** Producer's accuracy (PA) and user's accuracy (UA) are reported after applying post-classification methods. The columns represent the true classes and the rows represent the classifier's predictions. A single pixel was unclassified due to the post-classification improvement methods.

| Class | Spawning redds | Vegetation | Underwater rocks | Aquatic vegetation | Anthropogenic feature | Sediment | Total | UA (%) |
|---|---|---|---|---|---|---|---|---|
| Unclassified | 0 | 1 | 0 | 0 | 0 | 0 | 1 | |
| Spawning redds | **962** | 0 | 0 | 100 | 43 | 0 | 1105 | 87.06 |
| Vegetation | 0 | **925** | 30 | 0 | 0 | 132 | 1087 | 85.10 |
| Underwater rocks | 0 | 5 | **920** | 0 | 0 | 0 | 925 | 99.46 |
| Aquatic vegetation | 49 | 0 | 0 | **973** | 0 | 0 | 1022 | 95.21 |
| Anthropogenic feature | 0 | 78 | 0 | 0 | **1001** | 0 | 1079 | 92.77 |
| Sediment | 0 | 4 | 273 | 0 | 0 | **937** | 1214 | 77.18 |
| Total | 1011 | 1013 | 1223 | 1073 | 1044 | 1069 | 6433 | |
| PA (%) | 95.15 | 91.31 | 75.22 | 90.68 | 95.88 | 87.65 | | |

## Discussion

The aim of this study was to develop a standardised approach allowing for mapping salmonid spawning grounds that is easy to use and low-cost. The analysis of UAV-derived imagery in the contrasting case study areas situated in the subarctic region showed a successful application of a pixel-based classification methods that were able to identify the spawning area from RGB imagery with high accuracy. The study area in lake Thingvallavatn was selected based on a two-step selection process where we (i) located spawning redds from the shore, and (ii) verified that the observed redds are used for spawning through in-situ observations and video recordings of spawning activity, and noted the presence eggs to confirm successful spawning. Imagery of the confirmed spawning grounds taken from a height of 50 m was deemed suitable for classifying spawning redds after which fifteen training areas with a similar number of pixels were selected for each spectral class. Of all methods tested (S2 Table) the maximum likelihood and neural net classification methods showed highest producer's and user's accuracy. The two algorithms differed slightly in their performance: the neural net classification showed more noise in the deeper waters of lake Thingvallavatn (Fig 2C), and misclassified aquatic vegetation as spawning redds and sediment as vegetation in lake Ellidavatn (Fig 3C). Even though in our case the maximum likelihood classification method performed better, both methods should be considered when applying to other study systems. Another factor to consider is computational power: the run time on a basic, off-the-shelf laptop is about 20 seconds and 23 minutes per image for maximum likelihood and neural net, respectively. The application of the three post-classification methods improved the accuracy for most endmember classes in both study areas

**Table 4. Error matrix of the neural network classification in pixels for lake Ellidavatn.** Producer's accuracy (PA) and user's accuracy (UA) are reported after applying post-classification methods. The columns represent the true classes and the rows represent the classifier's predictions.

| Class | Spawning redds | Vegetation | Underwater rocks | Aquatic vegetation | Anthropogenic feature | Sediment | Total | UA (%) |
|---|---|---|---|---|---|---|---|---|
| Unclassified | 0 | 0 | 0 | 0 | 0 | 0 | 0 | |
| Spawning redds | **1009** | 0 | 0 | 114 | 506 | 0 | 1179 | 85.58 |
| Vegetation | 0 | **929** | 160 | 0 | 0 | 241 | 1330 | 69.85 |
| Underwater rocks | 0 | 0 | **943** | 0 | 0 | 42 | 985 | 95.74 |
| Aquatic vegetation | 2 | 0 | 0 | **619** | 0 | 85 | 706 | 87.68 |
| Anthropogenic feature | 0 | 84 | 3 | 0 | **988** | 0 | 1075 | 91.91 |
| Sediment | 0 | 0 | 117 | 340 | 0 | **701** | 1158 | 60.54 |
| Total | 1011 | 1013 | 1223 | 1073 | 1044 | 1069 | 6433 | |
| PA (%) | 99.80 | 91.71 | 77.11 | 57.69 | 94.64 | 65.58 | | |

(S3–S6 Tables). Most notably, in the lake Ellidavatn study area (S5 and S6 Tables) the PA of the spawning redds was improved by almost 3% showing the importance of post-classification improvement methods. However, the exact specifications of these methods will depend on its study area. For example, the irregular nature of the Arctic charr spawning grounds in lakes Thingvallavatn and Ellidavatn prompted the use of specific values (see methods). In the case of more regularly shaped spawning grounds a revaluation of these parameters would be recommended.

To map the area of entire spawning habitats using UAV and semi-automated processing has several benefits: (i) it takes less time when large quantities of images need to be analysed in case of frequent monitoring, (ii) improves accuracy in case of observer bias [24], and (iii) the combination of a single RGB image and supervised classification methods makes our approach accessible to a wide range of users. Rather than estimating the number of spawning redds, the method we are proposing here maps the spawning redds visible on a RGB image which will address some of the major shortcomings of the redd count method. For example, manual count is highly inaccurate in areas with superimposed and/or interconnected redds [24, 41]. The method presented here offers a new way of dealing with these shortcomings where instead of counting the number of redds the pixels comprising the spawning redds are quantified, allowing for comparable results between observers and over spatial and temporal scales. Spatial distribution, connectivity, and size of spawning habitat are important indicators of spawning habitat quality and have shown correlation with redd occupancy [42, 43]. Furthermore, our approach allows for studying of the dynamics of spawning habitats through semi-automated and therefore cost-effective remote sensing-supported monitoring procedures, *i.e.* spatially analysing the spawning habitats with regards to whether they are getting bigger, smaller or are relocating elsewhere. Assessing such dynamics is of utmost importance in times of rapid environmental change. For example, shrinking spawning areas have been associated with increased temperature and salinity [44] and shifts in spawning habitats are associated with temperature fluctuations [45, 46]. Although it has been previously shown that predicted usable habitat correlates with salmonid redd count [47], further research into the direct correlation between redd numbers and overall spawning habitat size would be recommended to be able to make use of existing redd count data.

While there are a lot of benefits to mapping spawning redds using semi-automated processing, there are some limitations to keep in mind. For example, good weather is needed to obtain the image quality necessary for data processing: high winds affect the stability of the drone and cause water surface disturbance; water turbidity and reflection of the sun can also be a problem. Obtaining data using a UAV may require more planning than traditional redd counting, but it is far more efficient. A limitation to the method is that suitable reference points are needed on the collected images. This is usually not an issue when spawning redds are located close to the shoreline, but can be more problematic when they are in open water. In such a case, a reference point such as a buoy or other clearly identifiable geographic marker would be needed. While the data for this study was collected by an experienced drone pilot, this is no longer necessary as drones are becoming more user-friendly. And while some practice to fly a UAV is recommended, current off-the-shelf drones are suitable as they often come with GPS, VPS (visual positioning system), and collision avoidance technology, making them very easy to use. Our results show that an off-the-shelf drone produces images that are suitable for mapping spawning redds. We used two different models (DJI Mavic Pro and DJI Mavic 2 Zoom) with different sensors and while the reported accuracy from Ellidavatn (DJI Mavic 2 Zoom) was slightly better, in both cases we report >90% accuracy (Tables 2–4). Finding the right camera settings may also take some practice. To ensure optimal image quality when data is collected by amateur drone pilots we recommend the use of automatic settings.

Large-scale monitoring programs are needed to assess ecosystem changes caused by climate change and other anthropogenic stressors. Our approach to detect spawning habitats has the potential for future use in complementing fieldwork that aims at monitoring changes in salmonid spawning grounds in different environmental conditions. Scaling this method to other freshwater ecosystems would allow to assess the full potential of this method in detecting spawning redds from a range of different species. We show the successful application of this method in two very contrasting lake environments which demonstrates its potential. Our method would furthermore allow to include the public in data collection through crowdsourcing as the ubiquitous existence of UAVs gives the opportunity to make use of crowdsourcing to gather images of salmonid spawning grounds. In principle an amateur drone pilot should be able to accomplish data collection without complicated technical guidance. Adopting the analysis to freely available open-source tools would help to even further advance this method to a low-cost approach that is accessible for regular monitoring programmes. For example, QGIS is a suitable open-source platform for which a Semi-Automatic Classification Plugin [48] has been developed that can be used with this pipeline. For further monitoring purposes, a framework building on the capabilities of Geographic Information Systems (GIS) will need to be developed to create a spatial database from the remote sensing analyses [49]. To achieve the objective of analysing regular observations in a consistent manner across space and time, it is essential to carefully consider a robust photogrammetric processing approach and identify suitable ground control points for reliable georectification of the data. This process lays the foundation for establishing change detection procedures, which play a crucial role in assessing spatial changes in spawning grounds over time. Additional geographical observations are also derived from the remote sensing analyses as basic indicators of the different endmember classes (*e.g.* aquatic vegetation, vegetation, shoreline) providing useful information for a GIS database. At the current stage of our case study presented here the additional endmember classes are only used to distinguish the spawning redds from other features. However, with further development of the analyses there is scope for the other endmember classes to be utilised for further ecological analyses. Such information will be valuable *e.g.* for geomorphological mapping of the study sites to further understand the ecosystem and see how its geography is changing beyond the nature of the nesting sites.

## Conclusions

The method we proposed here is a standardised method to map spawning redds that is easy in use and low-cost. To apply the method to other environments the following three steps are to be followed: (i) data acquisition, (ii) data processing, and (ii) accuracy assessment (Fig 4).

Data acquisition starts with selecting the study site and planning the drone mission based on suitable weather conditions. It is important that images are taken with minimum water surface reflection. This can be done by taking images when the sun is not at its highest point, and by conducting the survey at calm wind conditions to minimise ripples in the water surface. To acquire the images we recommend automatic camera settings and to further increase the contrast a polarising filter can be added. We also recommend taking RAW images which offer the highest resolution possible and the opportunity to improve the images during the image processing stage. Our results show that a UAV flight altitude of 50 m was best suited. It is imperative that the camera is turned at a 90˚ angle to take the images from straight above.

Data processing starts with selecting the endmember classes. Following this, the regions of interest (ROIs; training data) need to be selected for each endmember class, while keeping in mind that the amount of pixels per class should be similar. The next step involves applying the supervised classification algorithms (maximum likelihood, neural net). In our case, no

probability threshold was set to classify all pixels in the image for the maximum likelihood classification in *ENVI*. The parameters used to run the neural net classification in *ENVI* were set to logistic activation, 0.9 training threshold contribution, 0.2 training rate, 0.9 training momentum, 0.1 training RMS exit criteria, 1 hidden layer, and 1000 training iterations. Post-classification methods can be applied to improve the accuracy (majority-minority, analysis, sieve classes method, clump classes method).

The last step of the method covers the accuracy assessment for which ground truth data needs to be selected after which the confusion matrix based on ground truth ROIs is applied.

## Supporting information

**S1 Table. Technical specifications UAV.** Basic technical specifications of the RGB camera and battery carried by the DJI Mavic Pro and DJI Mavic 2 Zoom drone.
(PDF)

**S2 Table. Accuracy assessment spawning redd class in lake Thingvallavatn.** Accuracy assessment of the seven supervised image classification methods of the spawning redd class in lake Thingvallavatn. Producer's Accuracy (PA) and User's Accuracy (UA) are presented in percentages and pixels before post-classification methods have been applied.
Table furthermore reports overall accuracy (%) and the kappa coefficient (k).
(PDF)

**S3 Table. Accuracy assessment maximum likelihood lake Thingvallavatn.** Results of accuracy assessment of the maximum likelihood classification algorithm in lake Thingvallavatn before and after applying post-classification methods. Reported are producer's Accuracy (PA) and User's Accuracy (UA) by class.
(PDF)

**S4 Table. Accuracy assessment neural net lake Thingvallavatn.** Results of accuracy assessment of the neural network classification algorithm in lake Thingvallavatn before and after applying post-classification methods. Reported are producer's Accuracy (PA) and User's Accuracy (UA) by class.
(PDF)

**S5 Table. Accuracy assessment maximum likelihood lake Ellidavatn.** Results of accuracy assessment of the maximum likelihood classification algorithm in lake Ellidavatn before and after applying post-classification methods. Reported are producer's Accuracy (PA) and User's Accuracy (UA) by class.
(PDF)

**S6 Table. Accuracy assessment neural net lake Ellidavatn.** Results of accuracy assessment of the neural network classification algorithm in lake Ellidavatn before and after applying post-classification methods. Reported are producer's Accuracy (PA) and User's Accuracy (UA) by class.
(PDF)

## Acknowledgments

The authors wish to thank the authority of Thingvellir national park for flight permission over Ólafsdráttur. This work has been built upon preliminary works in a MS thesis project by Silvia García Martínez.

## Author Contributions

**Conceptualization:** Kalina H. Kapralova, Benjamin D. Hennig.

**Data curation:** Fredrik Holm.

**Formal analysis:** Lieke Ponsioen.

**Funding acquisition:** Benjamin D. Hennig.

**Methodology:** Kalina H. Kapralova, Benjamin D. Hennig.

**Supervision:** Kalina H. Kapralova, Benjamin D. Hennig.

**Writing – original draft:** Lieke Ponsioen.

**Writing – review & editing:** Lieke Ponsioen, Kalina H. Kapralova, Fredrik Holm, Benjamin D. Hennig.

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
