## [Decision Letter · Decision Letter 0]

22 Jun 2023

PONE-D-23-17918Remote sensing of salmonid spawning sites in freshwater ecosystems: The potential of low-cost UAV dataPLOS ONE

Dear Dr. Ponsioen,

Thank you for submitting your manuscript to PLOS ONE. After careful consideration, we feel that it has merit but does not fully meet PLOS ONE’s publication criteria as it currently stands. Therefore, we invite you to submit a revised version of the manuscript that addresses the points raised during the review process.

We look forward to receiving your revised manuscript.

Kind regards,

Benigno Elvira, Ph.D.

Academic Editor

PLOS ONE

Journal Requirements:

Reviewers' comments:

Reviewer's Responses to Questions

**Comments to the Author**

1. Is the manuscript technically sound, and do the data support the conclusions?

Reviewer #1: Yes

Reviewer #2: No

2. Has the statistical analysis been performed appropriately and rigorously? 

Reviewer #1: Yes

Reviewer #2: No

3. Have the authors made all data underlying the findings in their manuscript fully available?

Reviewer #1: Yes

Reviewer #2: Yes

4. Is the manuscript presented in an intelligible fashion and written in standard English?

Reviewer #1: Yes

Reviewer #2: Yes

5. Review Comments to the Author

Reviewer #1: The authors propose an easy low-cost approach to map entire spawning habitats using drones. I believe that this technic is useful for this kind of analysis, but have some limitations.

I believe that study is adequate, but I have some doubts about the validation methodology. I believe that it is necessary more detail about the determination of the extent of the spawning grounds. Why do you need to take embryos? How many embryos? What about scuba diving and video recording? Time effort and efficiency? Please, clarify it.

Finally, I believe that more discussion about the weaknesses and limitations of this technics is necessary. There are a lot of limitations to the use of drones to check fish spawning areas (water turbidity, similar and confusing images, drone accessibility, fish ecological behaviour, etc.) and this is worth to be discussed.

Finally, please, include the sentence “The data that support the findings of this study is available on the GeoVis Lab website at https://geovis.hi.is/research/data/” in the text, with the aim to facilitate access.

Line 19: Why “Although”? please, delete it.

Line 88: please, include the name and year of description of species: e.g. Salmo trutta (Linnaeus, 1758)

Line 91: why i.e.? this is a Latin abbreviation that minds something like “this is”. Please, delete it.

I believe that Table 1 is unnecessary. It offers information that can be included in the text.

Figures 3 and 4 can be mixed in a unique figure, this facilitates the comparison between images. The same to 5 and 6.

Reviewer #2: I read with interest the work by Ponsioen et al. entitled: "Remote sensing of salmonid spawning sites in freshwater ecosystems: The potential of low-cost UAV data". This paper uses lightweight unmanned aerial vehicles (UAVs) to monitor and map salmon spawning redd, avoiding time-consuming surveys by trained personnel. The work's rationale is interesting and useful in riverine ecology; however, the paper suffers several drawbacks that must be addressed before publication. The material and methods section lacks details to make this study applicable in other environments. The results only encompass the description of confusion matrices without addressing ecological or methodological aspects.

In my opinion, the paper is not suitable for publication in its current form. However, I recommend that the authors resubmit a new, improved version after addressing significant issues.

Specific comments

Introduction:

I suggest starting with the role of salmonids, stressing ecological aspects and their role as fisheries resources. Moreover, discussing the nursery role of specific habitats for setting redds, I suggest talking about how nurseries are defined and measured (See Beck et al., 2001, 2003). Considering other papers dealing with riverine mapping, the paragraph regarding using UAV-based imagery should be better developed. Since you are proposing a new approach capable of lowering the effort required for direct in situ sampling, I suggest also focusing on the other limitation (other than time) of traditional visual census methods, such as inter-observe variability, limited area and safety. Moreover, the superimposition issue should be better explained and discussed to highlight. Direct observation remains better for solving this issue than aerial imagery; therefore, this aspect should be clarified.

Lines 68-69: What do you mean by estimating "the entirely of spawning habitats"? The paper lacks any reference regarding such habitats' geographic extension and cover. In the results section, I suggest comparing the two surveyed sites considering the extension of these mapped habitats.

Line 72: Several times, you highlight that such monitoring can be done without using trained personnel usually working in the field. However, please consider the time and the trained operators needed for deploying and flying the drone other than the skill required for data processing and analysis.

Material and methods

The study areas are very well described; however, many of these characteristics are not essential in applying this method, so I suggest streamlining this part by moving, for instance, the description of species in the introduction. I also recommend adding an image of a redd taken from the ground, maybe underwater, to improve Fig. 1d.

Lines 126-130: The workflow in this section regarding the study area seems out of topic. It should be improved by adding more information in each part and inserting it at the end of the material and methods section to summarize all operative steps.

Data acquisition

Line 136: You mentioned that SCUBA diving had been carried out as part of the ground truthing, but in the paper, this part of data validation is missing, and all the confusion matrices have been built using only imagery interpretation. Why did you not use such information collected directly on the field as validation points for your classification?

Lines 138-141: Two different drones with different sensors (sensors size, resolution, ISO sensitivity, etc. )have been used for imagery acquisition, so it could be useful to discuss the best camera settings and accuracies of the results, also considering the quality of the acquired photo.

Line 156-157: How did you stitch together more than one image collected in a study area?

Results:

After discussing the best classification algorithm, I suggest adding some methodological and ecologic value results. Which is the best drone for such kind of acquisition and why? Camera settings and flight altitude are important for imagery acquisition, so some results should address these aspects. The two areas differ significantly in redds cover? If yes, why?

Is it unclear if a photogrammetric approach is used to generate the photomosaic of the first area. If so, how have the other cartographic products (DSMs) been validated? Can they serve to improve habitat classification? If you have not applied any kind of Structure from Motion (SfM) processing, the imagery used can be affected by significant errors in positioning. Regarding these last aspects, all the generated maps cannot be used as cartographic products being lacking in georeferencing (grids with corresponding coordinates) and orientation (north arrows).

Discussion

In the discussion section, some important considerations regarding the use of this data for population monitoring should be added. Open-source software can be used, but in your work, the pipeline proposed is based on ENVI software. The same results could be achieved also with QGIS? Could Wich limitations exist?

6. PLOS authors have the option to publish the peer review history of their article (what does this mean?). If published, this will include your full peer review and any attached files.

Reviewer #1: No

Reviewer #2: No

---

## [Author Response · Author response to Decision Letter 0]

3 Aug 2023

Journal Requirements:

We addressed the specific journal requirements highlighted in your letter, namely (1) we ensured that the manuscript meets PLOS ONE’s style requirements, (2) we have made our data and the pipeline available on GitHub via https://github.com/Worldmapper/ecoUAV-iceland, (3) we could not provide a grant number as this study was funded by The University of Iceland Research Fund awarded to Benjamin D. Hennig but these grants do not come with a grant number, (4) we have moved our data and the pipeline to a public repository (see above), and (5) we have moved our ethics statement into the method section.

Reviewers' comments:

Reviewer's Responses to Questions

Comments to the Author

1. Is the manuscript technically sound, and do the data support the conclusions?

Reviewer #1: Yes

Reviewer #2: No

2. Has the statistical analysis been performed appropriately and rigorously? 

Reviewer #1: Yes

Reviewer #2: No

3. Have the authors made all data underlying the findings in their manuscript fully available?

Reviewer #1: Yes

Reviewer #2: Yes

4. Is the manuscript presented in an intelligible fashion and written in standard English?

Reviewer #1: Yes

Reviewer #2: Yes

5. Review Comments to the Author

Reviewer #1: The authors propose an easy low-cost approach to map entire spawning habitats using drones. I believe that this technic is useful for this kind of analysis, but have some limitations.

I believe that study is adequate, but I have some doubts about the validation methodology. I believe that it is necessary more detail about the determination of the extent of the spawning grounds. Why do you need to take embryos? How many embryos? What about scuba diving and video recording? Time effort and efficiency? Please, clarify it.

Finally, I believe that more discussion about the weaknesses and limitations of this technics is necessary. There are a lot of limitations to the use of drones to check fish spawning areas (water turbidity, similar and confusing images, drone accessibility, fish ecological behaviour, etc.) and this is worth to be discussed.

Thank you very much for your feedback and helpful comments.

Confusion has arisen from the validation part of the text due to our poor wording. We have now adjusted our wording to make the steps clearer (115-119). We have taken out the mention of SCUBA diving as we did this to be able to collect the embryos which was done for another unrelated project. One of the steps is to note the presence of eggs to confirm successful spawning, but this can be done by a wide variety of ways, including snorkelling and SCUBA diving but more efficiently by wading through water using boots or waders. Even with SCUBA diving the validating of the spawning grounds took us about an hour.

We have added a paragraph to the discussion to address the limitations of using a drone to map spawning redds (372-391).

Finally, please, include the sentence “The data that support the findings of this study is available on the GeoVis Lab website at https://geovis.hi.is/research/data/” in the text, with the aim to facilitate access.

We have moved our data to a GitHub repository to make it more accessible and have added the sentence in results section (228-229).

Line 19: Why “Although”? please, delete it.

We removed it.

Line 88: please, include the name and year of description of species: e.g. Salmo trutta (Linnaeus, 1758)

This information has been added (91).

Line 91: why i.e.? this is a Latin abbreviation that minds something like “this is”. Please, delete it.

Removed.

I believe that Table 1 is unnecessary. It offers information that can be included in the text.

We have removed the table from the manuscript and any relevant information is mentioned within the text (104-108).

Figures 3 and 4 can be mixed in a unique figure, this facilitates the comparison between images. The same to 5 and 6.

We have done so.

Reviewer #2: I read with interest the work by Ponsioen et al. entitled: "Remote sensing of salmonid spawning sites in freshwater ecosystems: The potential of low-cost UAV data". This paper uses lightweight unmanned aerial vehicles (UAVs) to monitor and map salmon spawning redd, avoiding time-consuming surveys by trained personnel. The work's rationale is interesting and useful in riverine ecology; however, the paper suffers several drawbacks that must be addressed before publication. The material and methods section lacks details to make this study applicable in other environments. 

The results only encompass the description of confusion matrices without addressing ecological or methodological aspects.

In my opinion, the paper is not suitable for publication in its current form. However, I recommend that the authors resubmit a new, improved version after addressing significant issues.

We thank you for your thoughtful and detailed comments.

To show how our method can be applied to other environments we have adjusted the pipeline figure (Fig 4) and we have added the information to the conclusion (425-449).

We have moved methodological aspects to the results (218-255), this includes the selection of the drone flight altitude and the seven tested algorithms. We agree that adding ecological aspects is an important aspect, but we decided to focus on the spawning redds in this paper and selected the other endmember classes to be able to distinguish the spawning redds. To address ecological aspects further enhancement of the analyses is necessary and while possible, we believe it would bring a lengthy discussion and water down the main message we are trying to convey that it is an easy to use method to detect and map spawning redds. Instead, we have added in the discussion a sentence of the scope to classify other endmember classes in such a way that ecological analyses can be done (417-422). 

Specific comments

Introduction:

I suggest starting with the role of salmonids, stressing ecological aspects and their role as fisheries resources. Moreover, discussing the nursery role of specific habitats for setting redds, I suggest talking about how nurseries are defined and measured (See Beck et al., 2001, 2003). Considering other papers dealing with riverine mapping, the paragraph regarding using UAV-based imagery should be better developed. Since you are proposing a new approach capable of lowering the effort required for direct in situ sampling, I suggest also focusing on the other limitation (other than time) of traditional visual census methods, such as inter-observe variability, limited area and safety. Moreover, the superimposition issue should be better explained and discussed to highlight. Direct observation remains better for solving this issue than aerial imagery; therefore, this aspect should be clarified.

Thank you, we have now changed the introduction to reflect your suggestions (34-47).

We have also added a line in the discussion (355-359) to clarify the superimposition. 

Lines 68-69: What do you mean by estimating "the entirely of spawning habitats"? The paper lacks any reference regarding such habitats' geographic extension and cover. In the results section, I suggest comparing the two surveyed sites considering the extension of these mapped habitats.

We have revised the text and made it more clear (77-79).

As you point out in another comment the two spawning sites differ significantly in redd cover and redds differ in size, complexity and extent as well. So while the spawning grounds of lake Ellidavatn are not as complex and a single image covers all the redds, the spawning grounds in lake Thingvallavatn are highly complex and a composite image is needed to be able to detect all the redds. Therefore, we chose to use a single image for both of them to keep our results consistent. 

Line 72: Several times, you highlight that such monitoring can be done without using trained personnel usually working in the field. However, please consider the time and the trained operators needed for deploying and flying the drone other than the skill required for data processing and analysis.

We have added this to the discussion along with other weaknesses of using drones to map spawning redds (372-391). Some skill is indeed necessary but with time drones are getting more user friendly. New features are constantly being added such as GPS and collision avoidance technology. In our study we did use an experienced drone pilot to collect the data, but drones are constantly evolving making it nowadays possible for amateurs to use them. Every person on our team has now successfully collected data following our own pipeline.

Material and methods

The study areas are very well described; however, many of these characteristics are not essential in applying this method, so I suggest streamlining this part by moving, for instance, the description of species in the introduction. I also recommend adding an image of a redd taken from the ground, maybe underwater, to improve Fig. 1d.

We have streamlined the study area subsection (84-108). 

Following your recommendation we looked at different redd images taken underwater (see an example below). Neither of the underwater images examined showed a redd well enough. Thus, we decided to keep our current figure (fig 1d).

Lines 126-130: The workflow in this section regarding the study area seems out of topic. It should be improved by adding more information in each part and inserting it at the end of the material and methods section to summarize all operative steps.

We have removed a large part from the study area subsection (84-108). Furthermore, the pipeline is moved to the results (217), and following one of your previous comments the pipeline has been adjusted (Fig 4).

Data acquisition

Line 136: You mentioned that SCUBA diving had been carried out as part of the ground truthing, but in the paper, this part of data validation is missing, and all the confusion matrices have been built using only imagery interpretation. Why did you not use such information collected directly on the field as validation points for your classification?

Fair comment, this was also noticed by the other reviewer. Confusion has arisen from the validation part of the text due to our poor wording. We have now adjusted our wording to make the steps clearer (115-119). We have taken out the mention of SCUBA diving as we did this to be able to collect the embryos which was done for another unrelated project. One of the steps is to note the presence of eggs to confirm successful spawning, but this can be done by a wide variety of ways, including snorkelling and SCUBA diving but more efficiently by wading through water using boots or waders. Even with SCUBA diving the validating of the spawning grounds took us about an hour.

We also added (125-127) that the ground truthing was done on the day of data acquisition by visually confirming the presence of redds from the shore. We hope this clears it up.

Lines 138-141: Two different drones with different sensors (sensors size, resolution, ISO sensitivity, etc.) have been used for imagery acquisition, so it could be useful to discuss the best camera settings and accuracies of the results, also considering the quality of the acquired photo.

We have added this to the discussion (386-391).

Line 156-157: How did you stitch together more than one image collected in a study area?

We have clarified the text stating that a singular image was used (137-138).

Results:

After discussing the best classification algorithm, I suggest adding some methodological and ecologic value results. Which is the best drone for such kind of acquisition and why? Camera settings and flight altitude are important for imagery acquisition, so some results should address these aspects. The two areas differ significantly in redds cover? If yes, why?

We have moved a large part of our material and methods section to the results (218-255) as this encompassed methodological results (such as flight altitude selection and algorithm selection). Furthermore, we have added information about the camera settings to the material and methods (121-124) and discuss it in the discussion (386-391). 

And indeed the two spawning grounds differ in redds cover. This is due to lake Thingvallavatn having one major spawning ground and a short spawning time of two months while in lake Ellidavatn there are more locations with spawning grounds as well as a longer spawning time (September to the end of November). 

Is it unclear if a photogrammetric approach is used to generate the photomosaic of the first area. If so, how have the other cartographic products (DSMs) been validated? Can they serve to improve habitat classification? If you have not applied any kind of Structure from Motion (SfM) processing, the imagery used can be affected by significant errors in positioning. Regarding these last aspects, all the generated maps cannot be used as cartographic products being lacking in georeferencing (grids with corresponding coordinates) and orientation (north arrows).

In our study, the focus was not on generating a photomosaic using a photogrammetric approach. Therefore, no photogrammetric techniques were employed for this purpose since we worked with a single UAV image in the analyses. We apologise if this was not clearly conveyed in the article. The absence of a photomosaic was intentional as it fell outside the scope of our study objectives. The validation process involved rigorous cross-referencing with ground observations made during data collection at the particular location that was used in the analysis.

To clarify, we have added a paragraph where we acknowledge the needs for and benefits of using photogrammetric techniques as well as the importance of georeferencing for long term monitoring purposes (409-414).

Discussion

In the discussion section, some important considerations regarding the use of this data for population monitoring should be added. Open-source software can be used, but in your work, the pipeline proposed is based on ENVI software. The same results could be achieved also with QGIS? Could limitations exist?

We have added some more information on QGIS as this is a very suitable platform to which our pipeline can be transferred (405-407). A Semi-automated classification plugin has been developed for QGIS which is able to handle our pipeline.

---

## [Decision Letter · Decision Letter 1]

15 Aug 2023

Remote sensing of salmonid spawning sites in freshwater ecosystems: The potential of low-cost UAV data

PONE-D-23-17918R1

Dear Dr. Ponsioen,

We’re pleased to inform you that your manuscript has been judged scientifically suitable for publication and will be formally accepted for publication once it meets all outstanding technical requirements.

Kind regards,

Vitor Hugo Rodrigues Paiva, Ph.D.

Academic Editor

PLOS ONE

Additional Editor Comments (optional):

Reviewers' comments:

Reviewer's Responses to Questions

**Comments to the Author**

1. If the authors have adequately addressed your comments raised in a previous round of review and you feel that this manuscript is now acceptable for publication, you may indicate that here to bypass the “Comments to the Author” section, enter your conflict of interest statement in the “Confidential to Editor” section, and submit your "Accept" recommendation.

Reviewer #1: All comments have been addressed

Reviewer #2: All comments have been addressed

2. Is the manuscript technically sound, and do the data support the conclusions?

Reviewer #1: Yes

Reviewer #2: Partly

3. Has the statistical analysis been performed appropriately and rigorously? 

Reviewer #1: Yes

Reviewer #2: N/A

4. Have the authors made all data underlying the findings in their manuscript fully available?

Reviewer #1: Yes

Reviewer #2: Yes

5. Is the manuscript presented in an intelligible fashion and written in standard English?

Reviewer #1: Yes

Reviewer #2: Yes

6. Review Comments to the Author

Reviewer #1: All my comments have been addressed, in my opinion, this study is worthy to be published in the present form

Reviewer #2: (No Response)

7. PLOS authors have the option to publish the peer review history of their article (what does this mean?). If published, this will include your full peer review and any attached files.

Reviewer #1: No

Reviewer #2: No

---

## [Editor Report · Acceptance letter]

18 Aug 2023

PONE-D-23-17918R1 

Remote sensing of salmonid spawning sites in freshwater ecosystems: The potential of low-cost UAV data 

Dear Dr. Ponsioen:

I'm pleased to inform you that your manuscript has been deemed suitable for publication in PLOS ONE. Congratulations! Your manuscript is now with our production department. 

Kind regards, 

on behalf of

Dr. Vitor Hugo Rodrigues Paiva 

Academic Editor

PLOS ONE